

# Investigation of structural changes of atmospheric aerosol samples during two thermal-optical measurement procedures (EUSAAR2, NIOSH870)

Theresa Haller[1], Eva Sommer[1], Thomas Steinkogler[3], Christian Rentenberger[1], Anna Wonaschuetz[1], Anne Kasper-Giebl[3], Hinrich Grothe[2], Regina Hitzenberger[1]

[1]University of Vienna, Faculty of Physics, Vienna, 1090, Austria
[2]TU Wien, Institute of Materials Chemistry, Vienna, 1060, Austria
[3]TU Wien, Institute of Chemical Technologies and Analytics, Vienna, 1060, Austria

*Correspondence to:* theresa.haller@univie.ac.at

## Abstract

Thermal-optical measurement techniques are widely used for the monitoring of carbonaceous aerosols. Although results of different thermal-optical measurement techniques are comparable for total carbon, they can vary widely for values of elemental carbon especially in the presence of brown carbon. Charring of organic material during the inert heating phase of thermal-optical measurements was found to be a major confounder, but no literature about investigations of structural changes during this process in atmospheric aerosols is available. In a recent study we investigated these structural changes for combustion aerosol standard soot (CAST). Now we apply this approach to selected atmospheric aerosol filter samples and a subset of eight washed filter samples with low WSOCs loadings. To investigate structural changes, Raman spectra were obtained for samples heated to the corresponding temperature levels and gas atmospheres of the EUSAAR2 and NIOSH870 protocols. The temperature levels where changes in the Raman spectra occurred (i.e. changes in structure) varied for different samples. For the washed samples with low WSOC loadings and absence of other water soluble aerosol components such as inorganic salts, changes in structural ordering and darkening of the samples were not observed. For all samples, ion chromatography, integrating sphere measurements (yielding black and brown carbon data) and thermal-optical analyses were performed. We were able to show for the first time that the darkening of a sample (measured in terms of transmission laser signal) is not necessarily caused by an increase of structural ordering in the sample. Therefore we suggest that the widely used term "charring" should be used carefully when the darkening of a sample during thermal-optical measurement procedures is interpreted.

## 1 Introduction

Carbonaceous matter is an important component of atmospheric aerosols because of its large contribution to aerosol mass concentrations (20% - 40% in PM2.5 and up to 50% in PM10; Pöschl, 2005; Yttri et al., 2007) and because of adverse impacts on climate and human health (e.g. Bond and Bergstrom, 2006; IPCC, 2013; Highwood et al., 2006; Anderson et al.,





2012; Mesquita et al., 2017; EEA, 2017; Partanen et al., 2018; Rohr et al., 2016). Especially the light absorbing fraction of carbonaceous material (light absorbing carbon; LAC) has a large impact on radiative forcing due to its heating and/or cooling effects (e.g. IPCC, 2013). Depending on the composition and structure of the particles, carbonaceous particles can also have the capability to act as cloud condensation nuclei and ice nuclei (e.g. Spracklen et al., 2011; Pierce et al., 2007; Prenni et al., 2012; Häusler et al., 2018; Ikhenazene et al., 2020).

As carbonaceous aerosols are such an important fraction of the atmospheric aerosol, reliable measurement methods are necessary. Carbonaceous material can be classified optically into Brown Carbon (BrC) and Black Carbon (BC) or thermo-chemically into Organic Carbon (OC), Elemental Carbon (EC) and inorganic or Carbonate Carbon (CC) (for discussion of the terminology see e.g. Petzold et al., 2013). However, the definitions of the fractions are tied to the measurement methods more than to the actual internal structure of the material which makes reliable measurements challenging. Results for EC

and OC measured with different thermal-optical techniques (used routinely to monitor carbonaceous aerosols) suffer from biases by up to 44% for EC, while the values of total carbon (TC=EC+OC) are comparable within 5%-15% for different thermal-optical methods (e.g. Reisinger et al. 2008; Cavalli et al., 2010; Yu et al., 2002; Cheng et al., 2012; Hitzenberger et al., 2006; Panteliadis et al., 2015; Müller et al., 2004; Venkatachari et al., 2006; Watson et al., 2005; ten Brink et al., 2004). Although carbonaceous matter spans a rather continuous range from highly light absorbing graphitic-like material to non-

absorbing organic molecules (e.g. Pöschl, 2005), thermal-optical measurement techniques attempt to distinguish OC from EC: OC is defined as the thermally unstable organic part of the aerosol, which evolves in the absence of oxygen at temperatures below 550°C – 700°C (Bond and Bergstrom, 2006; Chow et al., 2004). EC is defined as the fraction in the aerosol which does not evolve below 550°C -700°C in the absence of oxygen and combusts in the presence of oxygen at temperatures above 600°C (Andreae and Gelencsér, 2006). OC and EC are separated in thermal-optical measurement

techniques by heating a filter sample stepwise first in an inert atmosphere and then in an oxidizing atmosphere. Different temperature protocols vary in height and duration of the single temperature steps, particularly in the maximum temperature of the inert phase (550°C - 900°C, Cavalli et al., 2010). The amount of carbon leaving the filter at each temperature plateau is measured. Filter samples often darken during the inert phase which is commonly explained by pyrolysis and/or charring, and this additional darker carbonaceous fraction is called "Pyrolyzed Carbon" (PC). The darkening is monitored with a

transmission and/or reflection laser signal. PC is more thermally refractive than its organic precursor and does not necessarily decompose in the inert phase. This fact is taken into account in the interpretation of thermal-optical analyses: Following the assumption that PC either evolves completely prior to the original EC or has the same optical and thermal properties as the original EC, a split point is set where the transmission or reflection signal reaches its initial value. The amount of carbon leaving the filter before the split point is assigned to OC and the amount of carbon leaving the filter after

the split point is assigned to EC. However, this optical correction is based on two essential assumptions which are not generally fulfilled: firstly PC does not evolve totally before EC, and secondly the specific light absorption coefficient is not equal for PC and EC – it is not even constant for PC during the whole heating procedure of one single thermal-optical



analysis (Yu et al., 2002; Chow et al., 2004; Subramanian et al., 2006; Han et al., 2007). As a consequence, the optical correction can lead to an over- or underestimation of EC or OC respectively.

The largest discrepancies between different thermal-optical methods were found for samples obtained under wintry conditions and the presence of biomass smoke, which leads to large amounts of BrC in the aerosol (Reisinger et al., 2008; Wonaschuetz et al., 2009, Cheng et al., 2012). BrC is a typical product of biomass combustion processes and is present in the atmospheric aerosol especially in winter e.g. in central and northern Europe (Lukács et al., 2007). It absorbs visible light weakly in the long wavelength range and stronger in the short wavelength range. Besides, it is more thermally refractive than

other organic matter and can therefore be seen as a substance with properties intermediate between those of elemental and organic carbon (e.g. Pöschl et al. 2003).

Extensive efforts were undertaken to refine the original thermal-optical measurement procedure (Birch and Cary, 1996) in order to achieve a more accurate setting of the split point (e.g. Cavalli et al., 2010; Chow et al., 2007; Cheng et al., 2012). Since pyrolysis was identified as a severe confounder (Yu et al., 2002; Cheng et al., 2012), several temperature protocols

were developed with the intent to reduce the formation of PC. The protocols differ in duration, temperature and number of the single heating steps and can therefore differ in the formation of PC and hence to different split points. The EUSAAR2 protocol (Cavalli et al., 2010) was defined as the standard method for European atmospheric aerosols (Brown et al., 2017) because it was found to produce less PC than other investigated protocols.

Atmospheric aerosols are highly complex mixtures of different carbonaceous and non-carbonaceous components. When a

sample is heated during the thermal-optical measurement procedure the material undergoes various chemical transformations, restructurings and interactions. Especially water soluble organic carbons (WSOCs) were found to be responsible for a large fraction of charring (13%-66%; Yu et al., 2002). Several inorganic constituents can influence charring in complex and sometimes contradictory ways: Ammonium bisulfate enhances the charring of starch and cellulose but reduces the charring of levoglucosan (Yu et al. 2002). Metal salts generally enhance the charring of OC at least in diesel

soot and ambient aerosols (Wang et al., 2010). As a consequence of this interplay of several complex chemical reactions there is no unequivocal effect of non-carbonaceous aerosol constituents on the charring behavior and consequently on the changes of the EC/OC split (Wang et al., 2010; Yu et al., 2002; Bladt et al., 2014). It is obvious that a simple prediction of the charring behavior of an atmospheric aerosol sample is not possible at the current state of knowledge. Besides, most of the studies about pyrolysis during thermal-optical measurement procedures are based on the analysis of the transmitted /

reflected laser signal (e.g. Cavalli et al., 2010; Cheng et al., 2012; Yu et al., 2002) or on the thermal properties of PC (Yu et al., 2002) and not on the actual structure of the formed material.

The terms "charring" and "pyrolysis" are often used interchangeably in the literature to describe processes which lead to a darkening of the filter sample with the sometimes implicit assumption that the darkening process is caused by the formation of a more structured phase of carbon, but usually explanations of underlying structural changes are absent. To our

knowledge no study to date has addressed the structural changes of an atmospheric aerosol sample during a thermal optical measurement procedure by analyzing graphitization or structural ordering within the material. In a previous study (Haller et





al., 2019) we investigated the increase of structural ordering of two widely different combustion aerosol standard soot types (produced with a CAST burner by propane combustion) during a thermal-optical heating procedure (NIOSH870). Using Raman spectroscopy, we found that structural ordering increased at 870°C when a BrC rich (and BC poor) sample was

heated in a He atmosphere, whereas a BC rich (and BrC poor) sample did not further change its already high degree of graphitization under these conditions.

This approach is now applied to the much more complex situation of atmospheric aerosols. For 21 atmospheric aerosol samples selected from a two-year set of daily filter samples (criteria see below) the changes of structural ordering during the inert phase of two thermal-optical measurement techniques were investigated. This was performed for original samples as

well as for a subset of samples that had been washed with water to remove WSOC and other soluble aerosol material. The NIOSH870 (Birch and Cary, 1996) and EUSAAR2 (Cavalli et al., 2010) protocols were chosen because of the different production of PC reported by Cavalli et al. (2010). A dual-optics thermal-optical analyzer (Sunset Instruments Inc.) was used to produce heated samples and to determine EC and OC. Samples heated to each temperature plateau of the inert phases of both temperature protocols were analyzed with Raman spectroscopy. This technique is sensitive to C-C bonding types and

provides information about the degree of ordering in the material. Several properties of the samples (e.g. ionic composition, EC/OC, BC/BrC) were compared with the progress of structural changes derived from the Raman spectra.

## 2 Experimental

### 2.1 Sampling

Atmospheric aerosol was sampled on quartz fiber filters (Pall Tissuquartz 2500 QAT-UP, 47mm) at the rooftop lab of the

physics building of the University of Vienna (35m above ground) located about 1.5 km from the city center. Although the building is located in an area highly impacted by traffic, the aerosol at the rooftop lab can be seen as an urban background, since the sampling inlet faces the enclosed source-free courtyard of the building. The aerosol was sampled with an automatic sequential filter sampler (SEQ47/50, Sven Leckel) equipped with a PM2.5 inlet and set to a flow rate of 2.3 m³/h. Sampling time was set to 23 h from 12:00 to 11:00 (UTC+1) of the  next day. The loaded filters were stored in the filter sampler for

about one to two weeks in a closed magazine before they were transferred to the lab and stored at -22°C in petri dishes closed with PARAFILM®. Before sampling the filters had been heated at 450°C for an hour to remove volatile organic substances and stored for at least 24h in a water-vapor-saturated atmosphere to prevent re-adsorption of OCs (Jankowski et al., 2008) and to ensure low OC blanks (blank values typically around 0.2 µgC/cm$^2$). The samples were collected in 2014 and 2015 for nearly every day from which we chose 21 samples for further analysis. The chosen filters were cut into aliquots

and some of these were washed to remove WSOCs and other soluble aerosol material, e. g. inorganic salts (see below). In the following discussions, aliquots of the untreated (i.e. unheated and unwashed) filters are referred to as "original" samples.





## 2.2 Sample selection

The samples were selected from the period 2014 and 2015 based on the air mass origins on the respective sampling days, the LAC (=BC + BrC) loadings and the amounts of BrC on the filter.

Air mass back trajectories were calculated with the HYSPLIT model (Draxler and Hess, 1997, 1998; Draxler, 1999) for all sampling days between January 2014 and December 2015. Daily 72 h back trajectories, starting at 48.00°N, 16.00°E (Vienna), at altitudes of 100, 300 and 500 m were calculated and clustered with the built-in clustering function of HYSPLIT. The clustering was performed for spring/summer (March-August) and fall/winter (September-February) separately, the number of clusters was chosen according to the automatic suggestion of the software, which gave 6 clusters each for summer

and winter. The clusters were assigned to air mass origins from the West and from the northern, eastern and southeastern sector respectively to cover diverse air masses. From these clusters, the days with the highest fraction (winter) and lowest fraction (summer) of BrC in LAC in combination with a generally high filter loading were identified. For these selected days more detailed back trajectories were calculated to verify that air mass trajectories were fairly constant over the whole sampling time of 23 h (altitudes: 100, 300, 500; starting times: 14:00, 19:00, 24:00, 05:00, 10:00 UTC+1). If this was the

case, the filter samples obtained at the respective days were chosen for further analysis.

## 2.3 Integrating sphere method

Light absorbing carbon (LAC) (i.e. BrC and BC) was analyzed for all filter samples with an extension (Wonaschuetz et al., 2009) of the original integrating-sphere technique (an extensive description is given by Hitzenberger and Tohno, 2001). Circular filter punches with diameters of 10 mm were immersed in a mixture of 10% isopropanol, 40% $H_2O$ and 50%

acetone in polyethylene (PE) vials. The immersion in this mixture reduces enhanced absorption caused by possible non-absorbing coatings of the particles: Soluble material is removed, and the effect of insoluble coatings is reduced due to the similar refractive indices of the liquid and the coatings (Hitzenberger and Tohno, 2001). The PE vials were placed in the center of a 6 inch integrating sphere coated internally with a highly diffusely reflective material (Spectraflect™). The sample was illuminated with a halogen light source equipped with a diffusor and three interference filters (450, 550 and 650 nm) and

the wavelength-dependent light signal was recorded with a photodiode.

The BC and BrC content in the sample was then calculated in an iterative procedure described by Wonaschuetz et al. (2009), comparing the wavelength-dependent light signals with calibration curves obtained with a proxy for BC (Elftex 124, Cabot Corp.) and a proxy for BrC (humic acid sodium salt, Acros Organics, 20 no. 68131044). The values for BrC should therefore be understood as "humic acid sodium salt equivalent" and refer to the total mass of BrC including other

atoms (humic acid sodium salt contains 47% carbon by mass). Detection limits are 1 μg per sample for BC and 10 μg per sample for BrC.



## 2.4 Thermal-optical measurements and sample preparation

The EC and OC content of the samples was measured with a dual-optics thermal-optical instrument (Sunset Instruments Inc.). In this instrument the sample is heated stepwise first in He, then in an oxidizing atmosphere (2% $O_2$ in He). The 
duration times of the steps and the temperatures and atmospheres for two different heating protocols (EUSAAR2 and NIOSH870) are listed in Table 1. The evolving carbon during each temperature step is tracked by a flame ionization detector (FID). The FID signal is proportional to the evolving carbon content and is calibrated by introducing a known amount of methane at the end of each filter measurement. The darkening of the sample during the heating procedure in the inert phase is monitored with a (transmission and reflection) laser signal (laser wavelength: 635 nm) which is used afterwards to correct 
the results for pyrolysis: A split point is defined when the laser signal reaches its initial value and the carbon evolving prior to the split point is assigned to OC and the carbon evolving afterwards to EC. The precision of the instrument is 6% for OC as well as for EC and the lower detection limit is 5 $\mu gC/cm^2$ for OC and 1 $\mu gC/cm^2$ for EC. The instrument evaluation software (calc415) gives OC and EC loadings for each temperature level.

Twelve out of the 21 atmospheric aerosol samples were analyzed with both EUSAAR2 and NIOSH870 protocols (Table 1), 
the other eight samples only with EUSAAR2, as parts of these filters were washed to remove water soluble material including WSOC (see below) and were therefore not available for extra analysis with another protocol. For both protocols heated samples were prepared corresponding to the temperature levels of the inert phase and the first oxidizing temperature level of EUSAAR2: Aliquots of the filters were heated in the instrument following the NIOSH870 and EUSAAR2 temperature steps and atmospheres, respectively, before the procedure was interrupted at the desired temperature level. All 
samples were cooled down in He to below 75°C, removed from the oven and stored in a closed petri dish at laboratory temperature.

| Carrier Gas | NIOSH870 | | | EUSAAR2 | | |
|---|---|---|---|---|---|---|
| | duration [s] | temperature [°C] | code | duration [s] | temperature [°C] | code |
| He | 80 | 310 | NI310 | 120 | 200 | EU200 |
| He | 80 | 475 | NI475 | 150 | 300 | EU300 |
| He | 80 | 615 | NI615 | 180 | 450 | EU450 |
| He | 110 | 870 | NI870 | 180 | 650 | EU650 |
| He | 45 | 550 | | 30 | - | |
| He+$O_2$ | 45 | 550 | | 120 | 500 | EU500 |
| He+$O_2$ | 45 | 625 | | 120 | 550 | |
| He+$O_2$ | 45 | 700 | | 70 | 700 | |



| He+O$_2$ | 45 | 775 | | 80 | 850 | |
| He+O$_2$ | 45 | 850 | | - | - | |

**Table 1: Duration times, temperatures and gas atmospheres for the two different thermal-optical measurement protocols (NIOSH870, EUSAAR2). For the temperature levels highlighted in grey, heated samples were prepared and analyzed with Raman spectroscopy (see below). The codes are used when we refer to samples heated to a certain temperature – symbolized by the numeric index – following NIOSH870 (NI) or EUSAAR2 (EU).**

## 2.5 Washing of the samples

Eight randomly selected filters (out of the 21 samples) were cut in half and one piece of each filter was immersed in 80 ml of
MilliQ water (18 MΩcm, Millipore) in 100 ml bottles and shaken mechanically in a shaking water bath (GFL Technology, type GFL 1086) for about 20 minutes at room temperature with 100 rpm to remove most of WSOC as well as soluble aerosol material (e.g. Gao et al., 2003; Kirchstetter and Novakov, 2004). The washed filters were removed from the bottles, transferred into petri dishes and dried in a desiccator for 3-4 days. The dry filters were then stored at -22°C in closed petri dishes.

## 2.6 Raman Spectroscopy

In this study a confocal Raman microscope (Horiba Jobin-Yvon LabRAM 800HR) was used. The Raman microscope was equipped with a 632.8 nm HeNe-laser (maximum output <20 mW), a 532 nm frequency-doubled NdYAG DPSS laser (Oxxius LMX-532, maximum output <52 mW) and a CCD camera (Peltier cooled to -60°C). The laser beam was focused onto the sample with a 20x magnification objective lens (CF Plan, 20x/0.35, WD 20.5 mm, Nikon) and a 100x long working
distance (LWD) objective lens (MPlanN, 100x/0.75, WD 0.21 mm, Olympus). The spectra were calibrated with the silicon peak at 520.07 cm$^{-1}$. Raman signals were obtained for 10 spots for each (unheated and heated) filter sample within an area of (1x2) mm$^2$. We found that this area was large enough to characterize a whole sample. The acquisition times were chosen between 0.005 and 3 seconds and 30 – 3000 single spectra were accumulated for each position (slit 400 μm, confocal hole 500 μm, laser power between 25% and 100%). The particular settings (acquisition time, number of accumulations, laser
power) were chosen individually to obtain the best signal to noise ratio for each sample, since the fluorescence background intensity was different for the samples.

Due to instrument availability all samples were measured using the HeNe (red) laser except the heated and unheated samples of the washed filters (see below) and their unwashed counterparts which were measured using the NdYAG (green) laser. For samples heated to 500°C in O$_2$+He (EUSAAR2) and 870°C in He (NIOSH870), which were measured using the red laser, a
100x LWD objective lens and a smaller confocal hole (3 μm) were used to focus on single particles since the red laser in combination with the 20x objective lens was not sufficient to obtain spectra for the small amounts of material left on the filters at these heating stages. The laser power was decreased to 1% and accumulations of 400 single spectra and an



acquisition time of 0.5 sec were required. Because of these restrictions, Raman spectra were obtained for only five different

spots for these samples. Mean spectra and their standard deviations were calculated for the spectra of all samples.

Typical Raman spectra of soot and related carbonaceous material show two overlapping peaks at about 1350 cm$^{-1}$ (D-peak or

defect peak) and 1580 cm$^{-1}$ (G-peak or graphitic peak) (e.g. Fig. 1). Detailed descriptions are given by e.g. Rosen and

Novakov, 1978; Sadezky et al., 2005; Ferrari and Robertson, 2000; Schmid et al., 2011; Knauer et al., 2009; Ivleva et al.,

2007. The ratio of the D and G peak intensities (D/G) is an indication of the degree of structural ordering in the

carbonaceous material (Tuinstra and Koenig, 1970; Ferrari and Robertson, 2000). For small crystallite sizes < 2 nm as can be

found in combustion soot particles (see e.g. Haller et al., 2019), D/G is proportional to the probability to find 6-fold aromatic

rings in the cluster, which means that an increasing D/G ratio indicates increasing crystallite sizes (Ferrari and Robertson,

2000; Zickler et al., 2006) and therefore an increase of structural ordering. A more detailed description of the interpretation

of Raman spectra of soot can be found in the studies by Ferrari and Robertson (2000) or Haller et al. (2019). Several other

authors use this interpretation of the D/G ratio for Raman spectra of combustion aerosol samples (e.g. Commodo et al., 2016;

Ess et al., 2016) and use a higher D/G ratio as an indication of a higher degree of structural ordering within the material.

**2.7 Electron Microscopy**

As crystallite sizes are important for the interpretation of the D/G ratio in the Raman spectra, the crystallite sizes of three

samples (16.11.2014, 09.06.2015, 10.11.2015) were exemplarily determined from electron diffraction patterns obtained with

an electron microscope. A 200 kV transmission electron microscope (TEM) Philips CM200 equipped with a Gatan$^{TM}$ Orius

CCD camera was used to obtain electron diffraction patterns. Intensity profiles of the diffraction patterns were obtained by

both azimuthal integration along rings and background correction using PASAD-tools (Gammer et al., 2010). By fitting

pseudo-Voigt functions to the intensity profiles taken from at least 6 different locations on each sample, the average full

width at half maximum of the first intense diffraction peak was determined and used to calculate the mean crystallite size.

**2.8 Ion chromatography**

Aliquots of the filters (areas between 78 and 101 mm$^2$ depending on filter loadings) were extracted in 2 mL of ultrapure

water (MilliQ, 18 MΩcm, Millipore). After ultrasonic agitation for 20 minutes and centrifugation for 5 minutes at

13400 rpm, the extracts were analyzed by suppressed ion chromatography. For the determination of the anions (Cl$^-$, NO$_2^-$,

NO$_3^-$ and SO$_4^{2-}$) a Dionex Aquion system (Thermo Fisher) equipped with CS16 and CG16 columns, a CERS 500 self-

regenerating suppressor and a DS6 heated conductivity detector was used. The cations (Na$^+$, NH$_4^+$, Mg$^{2+}$, Ca$^{2+}$ and K$^+$) were

determined with a Dionex ICS-1100 system (Thermo Fisher) equipped with AS22 and AG22 columns, an AERS 500 self-

regenerating suppressor and a DS6 heated conductivity detector. The calibration was performed with external standards,

prepared from Merck stock solutions with 1000 ppm. The limit of detection for Cl$^-$ is 0.005 mg/L, for all other analytes

0.01 mg/L.


## 3 Results

### 3.1 Properties of the original (unwashed and unheated) samples

The restructuring of carbonaceous material was investigated with special emphasis on the presence of BrC. Therefore, winter samples with high BrC loadings and high BrC/LAC ratios and summer samples with BrC loadings below detection limit were selected for comparison purposes. The seven samples from summer 2015 had LAC loadings between about 3 and 5 $\mu g/cm^2$ with BrC/LAC = 0. The fourteen samples from winter 2014/2015 had LAC loadings between about 8 and 27 $\mu g/cm^2$ and BrC/LAC ratios from about 0.14 to 0.25 which were the highest BrC/LAC ratios in this period. TC values varied from 7 to 34 $\mu g/cm^2$ for summer samples and from 13 to 71 $\mu g/cm^2$ for winter samples.

The Raman spectra of all samples (e.g. Fig. 1, 2, 3) show two distinct peaks around 1350 $cm^{-1}$ (D-peak or defect peak) and 1580 $cm^{-1}$ (G-peak or graphitic peak) which is typical for soot. The Raman spectra of the original samples are surprisingly similar (Fig. 1), although the samples differ in composition and air mass origin. This could be explained by an appreciably higher contribution of graphitic like material (e.g. BC, EC) to the Raman signal in comparison to the contribution of possibly coexisting organic or brown carbon.

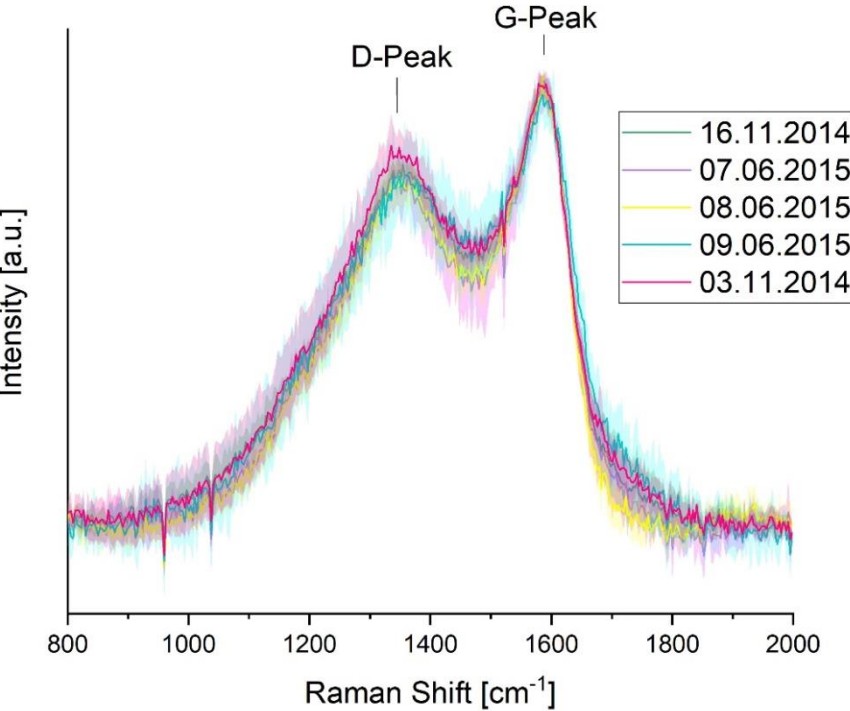

**Figure 1: Raman spectra for a representative subset of original (unheated, unwashed) samples. The spectra are similar among themselves within the error bars (shaded areas in the graphic) especially in the peak heights. For a better illustration, only five spectra were selected. All other spectra of original samples are comparable within error bars.**



Since the fraction of BC in PM2.5 was relatively constant over the whole year in our samples (Sommer, 2020), we conclude that traffic emissions (mainly by diesel cars) are the dominant source for BC even in wintertime when biomass fuels are burnt for space heating purposes.   The relatively constant source of diesel soot could lead to a relatively homogeneous and constant contribution of graphitic like species to the ambient aerosol over the whole year which in turn leads to similar Raman spectra of all original samples.

Crystallite sizes of the three samples analyzed by TEM are well below 2 nm (mean crystallite size of all three samples: 1.13 ± 0.1 nm). These sizes match with the crystallite sizes of two CAST soots in our recent publication (recalculated: 1.2 ± 0.1 nm for a BC rich and 0.8 ± 0.1 nm for a BrC rich sample; Haller et al., 2019). We therefore assume that also the crystallite sizes of the other atmospheric aerosol samples in this study should be below 2 nm.

## 3.2 Analysis of the unwashed (heated and original) samples

### 3.2.1 Comparison of structural changes EUSAAR2 vs. NIOSH870 after the inert phase

A comparison between NIOSH870 and EUSAAR2 protocols was performed for a subset of 12 unwashed samples (10 winter, 2 summer). Raman spectra were obtained for samples heated according to both the NIOSH870 and the EUSAAR2 protocol. A relative increase of the D-peak during the course of the inert phase occurs for both protocols (EUSAAR2 and NIOSH870). However, for some samples the total increase at the end of the He phase is larger for NIOSH870 than for EUSAAR2.

For four samples out of 12, the relative increase of the D-peak intensity at the maximum He temperature is clearly larger for NIOSH870 than for EUSAAR2, i.e. the error bars of NI870 and EU650 do not overlap (Fig 2). For two samples among these even the relative D-peak intensity of EU500 (i.e. the first temperature step in He+$O_2$) remains clearly lower than that of NI870 without an overlap of the error bars. For other four samples the difference between EU650 and NI870 is also observed, but the error bars overlap (in most cases due to a strong noise signal for NI870 resulting from low particle loadings of these samples). Only in three cases out of 12 the relative D-peak intensities are similar for NI870 and EU650. For one sample it was not possible to measure NI870 because of a too low mass loading after heat treatment.





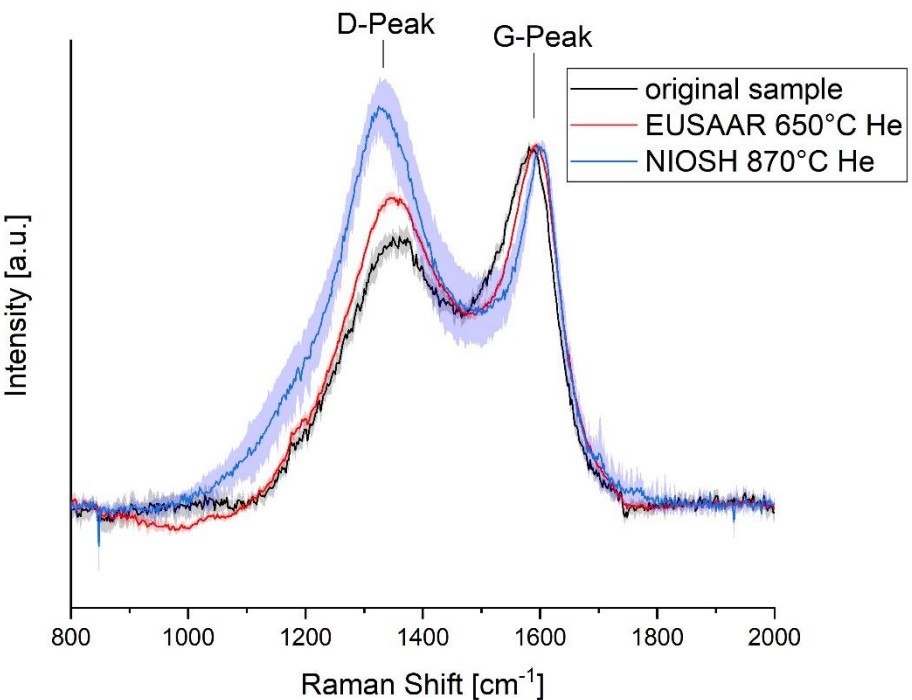

**Figure 2: Comparison of Raman spectra of EU650 (red line) and NI870 (blue line) for a representative sample (25.10.2014). The relative D-peaks increase in both cases compared to the spectrum of the original sample (black line) but the D-peak of NI870 is visibly higher than the D-peak of EU650.**

Since the crystallite sizes in the samples are < 2 nm, the interpretation by Ferrari and Robertson can be applied to the Raman spectra: A higher D/G ratio indicates a higher degree of structural ordering in the material.

The increase of the relative D-peak intensity for NI870 for some samples can therefore be interpreted as a stronger increase of structural ordering, when these samples are heated according to the NIOSH870 protocol compared to EUSAAR2. These differences indicate that PC which develops during the NIOSH870 protocol differs from the material developing during the EUSAAR2 protocol at least in four cases out of twelve. PC developed during the heating of these samples with different protocols might therefore also have different optical properties which would influence the progress of the transmission / reflection laser signal during the thermal-optical measurement procedure. A larger decrease of the laser transmission /reflection signal does therefore not necessarily indicate that more PC is formed. Several authors use the change of the transmission signal in combination with a fixed specific light absorption coefficient (e.g. 45 m$^2$/gC by Cavalli et al., 2010) to calculate the amount of PC formed during the thermal optical measurement procedure (e.g. Subramanian et al., 2007; Cavalli et al., 2010). Cavalli et al. (2010) conclude from a stronger decrease of the transmission laser signal during NIOSH-like protocols than during the EUSAAR2 protocol that more PC is built when the sample is heated following NIOSH-like protocols. However, our comparison of the Raman spectra of the heated samples of both protocols shows that this direct





inference from the transmission laser signal to the amount of PC may not generally be possible for atmospheric aerosol samples since the structure of PC and therefore also the optical properties are not necessarily the same in both cases. This finding is in good agreement with Yu et al. (2002), who found that light absorption coefficients of PC formed during a thermal-optical heating procedure are not identical.

### 3.2.2 Structural changes during EUSAAR2

For all of the 21 (unwashed) samples Raman spectra were taken for unheated samples and for samples heated according to the EUSAAR2 protocol. In most cases (17 out of 21) the relative intensity of the D-peak increases during the heating process indicating an increase of structural ordering. However, distinct changes of the Raman spectra occur at different temperature levels during the heating procedure in the inert phase (450°C – 650°C) or the oxidizing phase (500°C). Moreover in some cases the change happens only at one temperature level or extends over two temperature levels. Therefore the samples were classified into six categories (subsequently referred to as "Raman categories") depending on the temperature levels where the changes in the Raman signals occur: (Sample codes are explained in Table 1)

*Early structural change*: D-Peak of the Raman spectrum obtained for EU450 is distinctly higher than D-Peak of the spectrum for EU300 (i.e. error bars of spectra for EU450 and EU300 do not overlap) and D-Peak of EU650 is distinctly higher than D-Peak of EU450. The Raman spectra of the unheated sample, EU200 and EU300 have overlapping error bars as well as of EU650 and EU500. (Fig. 3, a)

*Late structural change:* D-Peak of the Raman spectrum of EU650 is distinctly higher than D-Peak of the Raman spectrum of EU450 and D-Peak of the Raman spectrum of EU500 is distinctly higher than D-Peak of the Raman spectrum of EU650. The spectra of the unheated sample, EU200, EU300, EU450 and EU650 have overlapping error bars. (Fig. 3, b)

*Fast and early structural change:* Only D-Peaks of the Raman spectra of EU300 and EU450 differ without overlapping error bars. Error bars of the spectra of the unheated sample, EU200 and EU300 overlap as well as error bars of the spectra of EU450, EU650 and EU500. (Fig. 3, c)

*Fast and late structural change:* Only D-Peaks of the Raman spectra of EU450 and EU650 differ without overlapping error bars. Error bars of the spectra of the unheated sample, EU200, EU300 and EU450 overlap as well as error bars of the spectra of EU650 and EU500. (Fig. 3, d)

*No change:* All mean spectra lie within the error bars of the other spectra. (Fig. 3, e)

*Undefined:* Changes of the Raman spectra cannot be assigned unambiguously to one of the other five categories as a consequence of too strong noise signals.





**Figure 3: Examples for the Raman categories "early structural change" (a), "late structural change" (b), "early and fast structural change" (c), „late and fast structural change" (d) and "no structural change" (e).**



Five samples each were assigned to the categories "early change" and "late and fast change", two samples each to "early and fast change" and "late change", four samples to "no change" and three samples to "undefined". The categorization was undertaken only for Raman spectra of samples heated following the EUSAAR2 protocol since this is the commonly used method in Europe.

330

It is remarkable that the spectra of the heated samples differ and show different progresses over the heating procedure (Fig. 3) although the Raman spectra of the original (unheated) samples are comparable. Therefore we conclude that the differences in the progress of restructuring might not be caused by the degree of structural ordering of the original samples or at least cannot be simply predicted by their Raman spectra.

335

### 3.2.3. Structural changes and laser signals (EUSAAR2)

During the thermal-optical analysis procedure with EUSAAR2 as well as with NIOSH870 a decreasing (transmission and reflection) laser signal was observed during the inert phases for all unwashed samples. The signals start to increase again during the last temperature level in He for both protocols. For a more detailed analysis of the progress of the laser signals we focus here on EUSAAR2. During the heating following EUSAAR2, the reflection laser signals decrease in more or less

340

pronounced steps at each temperature level in the He phase and no distinct differences between the samples are visible. No relationship between the changes in the Raman spectra and the (refl) laser signal is visible.

For the transmission signals different behaviors can be observed and the samples can be assigned to two categories regarding the transmission signals: a) a pronounced step in the (trans) laser signal at the second temperature level in He (300°C) (Fig. 4) and b) a relatively steady decrease of the (trans) laser signal during the first three temperature levels in He (up to 450°C)

345

(Fig. 5. The laser signals shown in Fig. 9 and 10 are color coded to identify the different samples, and the categories according to the changes in Raman spectra are identified by line type. No relationship between changes in the Raman spectra and the progress of the laser signal during heating is visible.

So the temporal changes of the reflection as well as the transmission laser signals can be similar although the Raman spectra of the respective samples indicate that an increase of structural ordering occurs at different temperature levels. Moreover, the

350

strongest decreases in the transmission laser signals (i.e. the steps in Fig. 9) do not occur at the temperature levels (450°C, 650°C), where changes in the Raman spectra are observable.



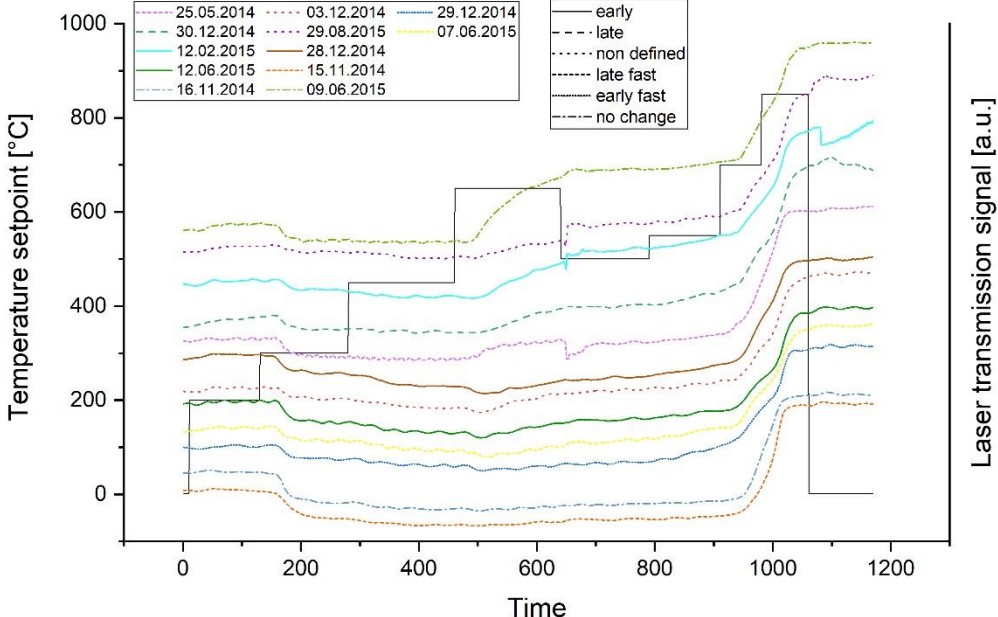

**Figure 4: Transmission laser signals during the heating procedure of the EUSAAR2 protocol for all cases with a pronounced step during the 300°C temperature level in He. The different line patterns represent the Raman categories early, late, late and fast, early and fast, no change and non-defined. The black solid line represents the ideal temperature curve in the instrument.**





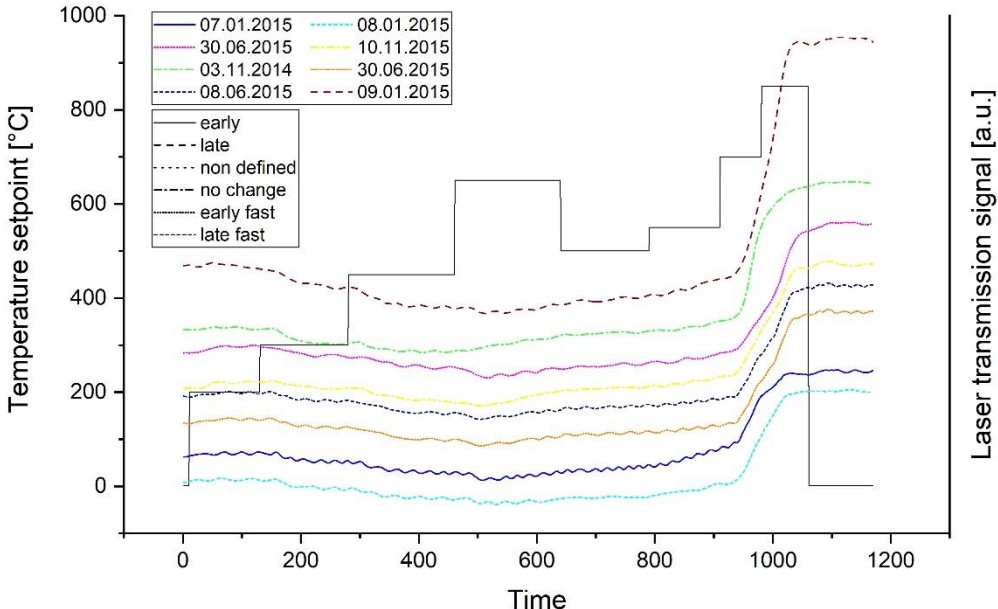

**Figure 5: Transmission laser signals during the heating procedure of the EUSAAR2 protocol for all cases with a steady decrease during the first three temperature steps (200°C – 450°C) in He. The different line patterns represent the Raman categories early, late, late and fast, early and fast, no change and non-defined. The black solid line represents the ideal temperature curve in the instrument.**

As the laser signals do not change appreciably at the same temperature levels where the Raman signals of the respective heated samples change, the darkening of the sample in terms of the red (trans, refl) laser signal does not correlate with changes in structural ordering. This is most striking for the four samples with no change in the Raman spectra of the heated samples (dashed-dotted lines in Fig. 4 and 5). Although the Raman spectra do not indicate an increase of structural ordering, the (trans, refl) laser signals decrease during the heating procedure in He, so the samples become darker without an increase of structural ordering, i. e. without a measurable graphitization of the material. We found similar effects in our earlier study (Haller et al., 2019) where we compared UV-VIS spectra of heated (NIOSH870 protocol) CAST soot samples with the changes of the respective Raman spectra: The UV-VIS spectra started to change at lower temperatures while the Raman spectra changed only at the highest temperature in He (870°C).

Therefore we conclude that a graphitization of the material cannot be the only reason for the darkening (i.e. stronger absorption of the red laser signal) of a sample. There must be other processes during heating in the inert phase, which form non-graphitic material with smaller optical band gaps than the unheated material and which therefore can absorb more strongly at the wavelength of the laser (635 nm) compared to the original material. Yu et al. (2002) assume that light absorbing intermediate OC products are formed during the inert phase of the thermal-optical measurement procedures they used (a NIOSH method and other experimental temperature protocols) which leave the filter already during the highest





temperature step in He. The increase of the laser signals starting at 650°C in He could therefore partly be explained by the decomposition of dark intermediate OC products without graphitic structures.

### 3.2.4. Possible relations of structural changes with other parameters

As there are often references to properties of the aerosol (e.g. Wang et al., 2010; Yu et al., 2002; Reisinger et al., 2008;
Wonaschuetz et al., 2009) and their influence on the charring behavior of samples or on the reliability of thermal-optical measurements, we investigated some of these possible effects using the unwashed samples. We compared the progress of structural changes (i.e. the assignment to the Raman categories) to the chemical compositions of the non-carbonaceous matter and to the compositions of the carbonaceous matter (LAC/BC, EC/OC).

The ionic compositions of the samples sorted by their Raman categories are shown in Fig. 6. No relationship between the
ionic components and the progress of restructuring of the material is visible.

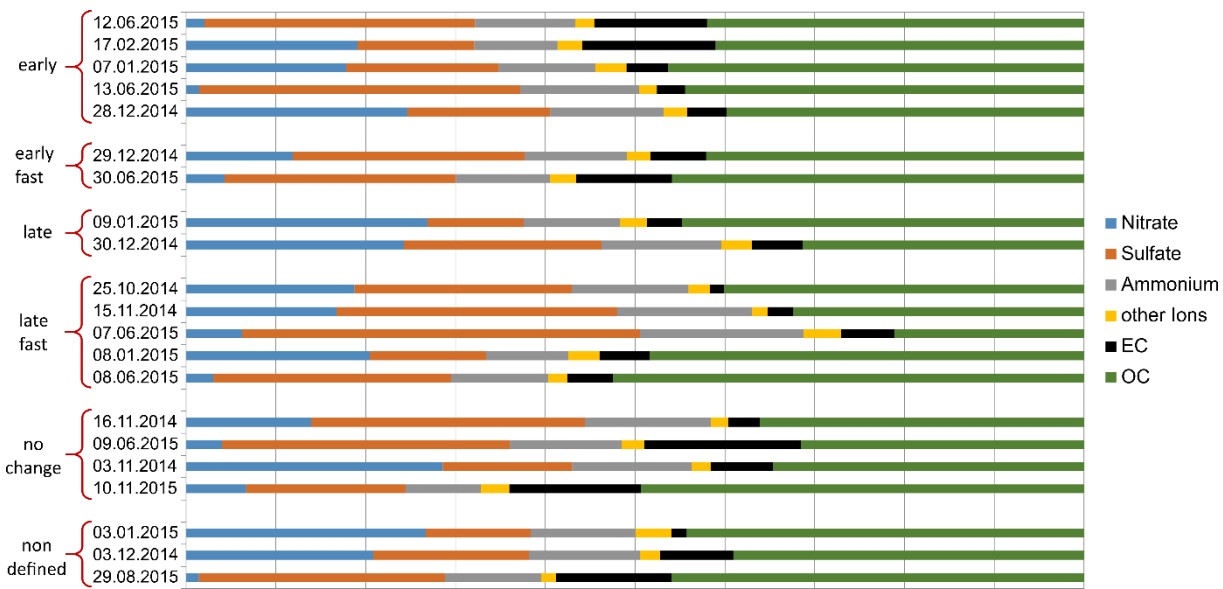

**Figure 6: Fractions of analyzed ions, elemental carbon (EC) and organic carbon (OC) sorted by the categories representing the behavior of the Raman spectra during the thermal optical heating procedure. The contribution of minor ions to the category**
**"other ions" can be found in the supplement.**

This could be explained by the study by Yu et al. (2002) who showed that one and the same compound can enhance or reduce charring depending on the organic substances present in the sample. Although they analyzed only three standards for organic carbon, it is possible that similar counteracting influences could be found also for other organic substances present in
atmospheric aerosols which could suppress a relationship between charring and the composition of the inorganic salts within the sample.





The progress of structural changes is also unrelated to the BrC/LAC ratio (Fig. 6). Samples with BrC below detection limit are found in nearly all the categories. Especially the samples in the category with no change in the Raman signal during 400 heating do not have appreciably lower BrC/LAC ratios. This fact is interesting, because previous studies in the Austrian aerosol (e.g. Reisinger et al., 2008; Wonaschuetz et al., 2009) found larger discrepancies between EC values obtained with different thermal-optical methods for atmospheric aerosol samples with high contributions of BrC to LAC, up to about 0.9 (Wonaschuetz et al., 2009) and 0.75 (Reisinger et al., 2008) (These values are comparable to BrC/LAC about 0.7 in this study). At the time of the earlier studies it was not possible to determine whether these discrepancies were due to a higher 405 tendency of BrC to char (and therefore influence the EC/OC split) or from other BrC-related processes. Our findings suggest that the relative amount of BrC has no influence on the increase of ordering during the heating procedure at least for the samples analyzed here. The discrepancies between different thermal-optical methods for BrC-rich samples might therefore have other reasons than a restructuring of BrC in terms of increased structural ordering.

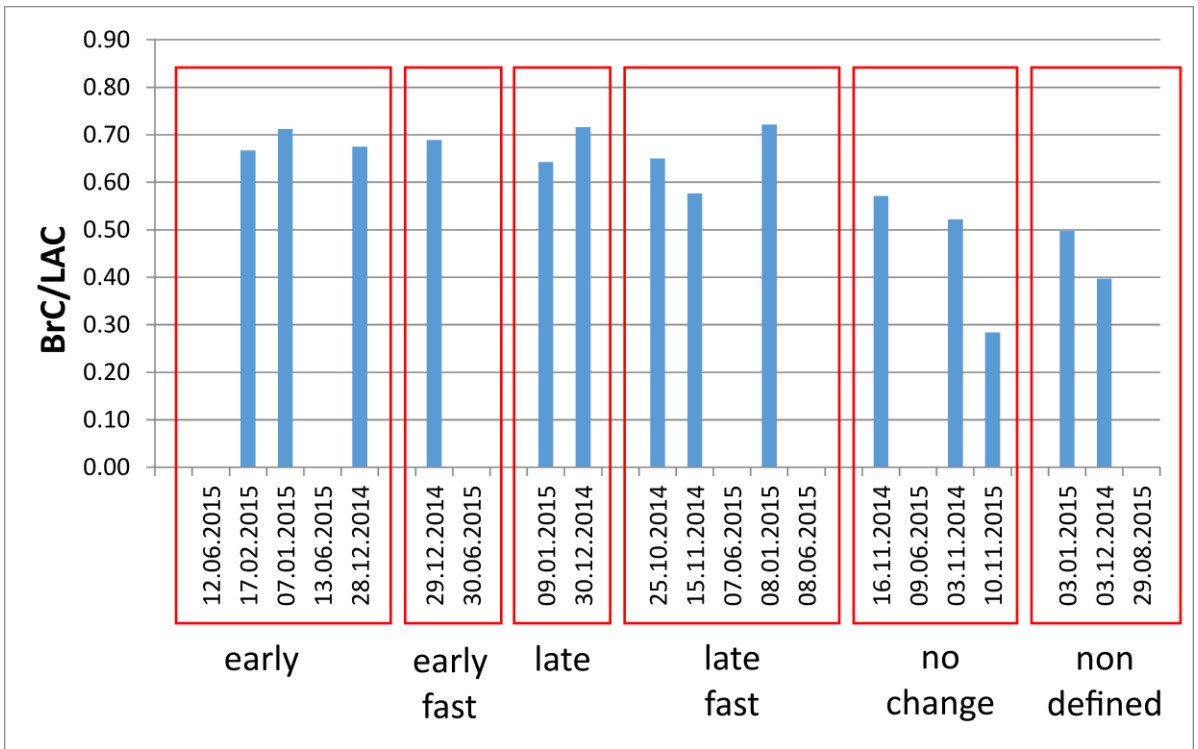

**Figure 7: Comparison of BrC/LAC with the categories representing the behavior of the Raman signal during the heating procedure of the thermal-optical analysis. For samples with BrC/LAC=0 (i.e. no bar in the graphic) BrC was below detection limit (<12.7 µg/cm²)**





### 3.5 Structural changes of the washed samples

Figure 8 shows two typical thermograms (EUSAAR2 protocol) one of an unwashed and one of a washed part of the same filter. While the transmission laser signal of the unwashed sample decreases during the inert phase, the signal of the washed sample is nearly constant until it starts to increase during the 650°C step in He. The initial laser signal for the washed sample is higher than the one for the original sample indicating that part of the light absorbing material was removed during the washing procedure. Integrating sphere measurements of the washed samples confirm that part of BC is absent and BrC is

below detection limit.

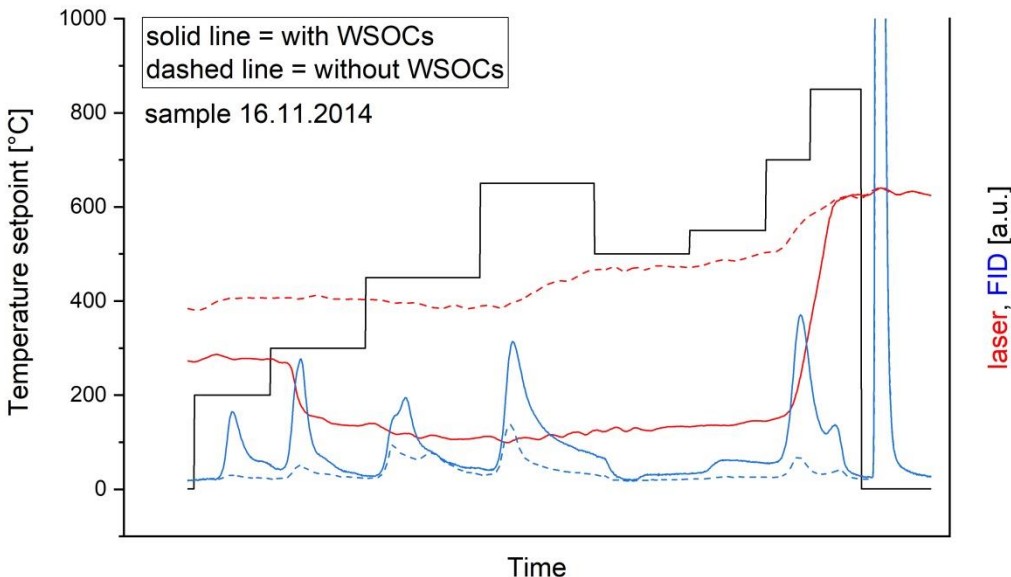

**Figure 8: Typical thermogram of a sample with (solid lines) and without (dashed lines) WSOCs. The transmission laser signal decreases during the first three temperature levels for the unwashed sample (red solid line) but stays nearly constant for the washed sample (red dashed line). The FID signals are lower for all temperature levels for the washed sample (blue dashed line)**
**than for the unwashed sample (blue solid line). The black solid line represents the ideal progress of the sample temperature.**

The differences in the laser signals between original and washed samples hold for all of the 8 investigated samples (Fig. 9). These findings are in good agreement with Yu et al. (2002) who found that WSOCs account for a large contribution (13-66%) to the charring during NIOSH-like heating procedures.

The Raman spectra of all washed samples do not change noticeably during the heating process although some of their unwashed counterparts have changing Raman signals when they are heated (Fig. 10) indicating that no increase of structural ordering occurs during heating of the washed samples. From the Raman spectra we conclude that the washed samples indeed





do not change their structure in terms of graphitization. In contrast to cases, where the laser signal decreases (as described in 3.2.1) it seems to be possible to conclude from an unchanged laser signal  to a lack of  charring or graphitization of the

material.

The laser signals start to increase during the last temperature step in He (650°C) also for the washed samples. Since in these cases no noticeable pyrolysis occurs, we assume that some of the light absorbing material which was originally on the filter evolves in the absence of oxygen at 650°C. This could be caused by a premature evolution of EC or the evolution of the water-insoluble fraction of BrC.

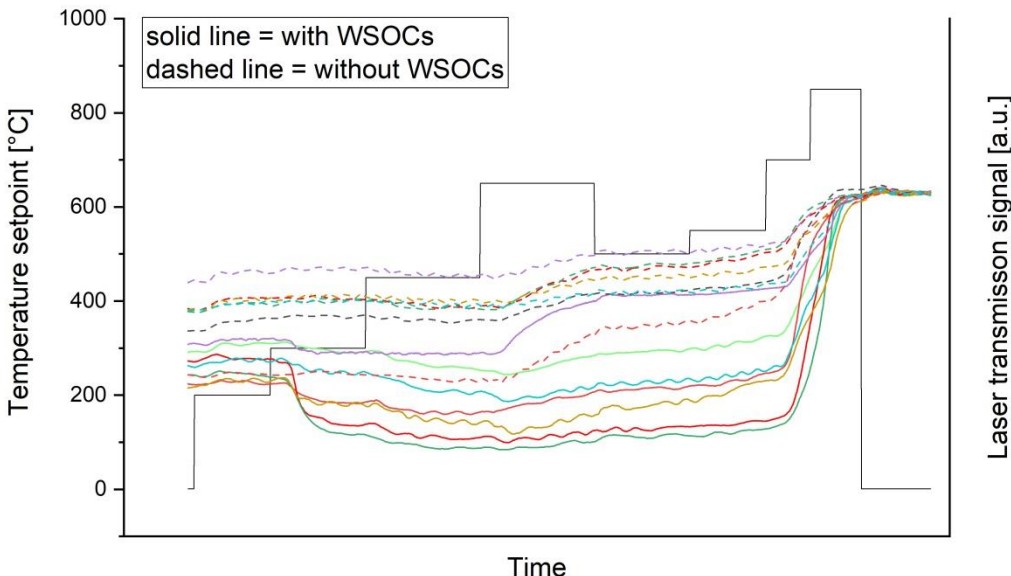


**Figure 9: Progress of the laser signals for all samples with and without WSOCs measured with the EUSAAR protocol. The laser-signals are normalized at the signal of the clean filter at the end of the procedure.**



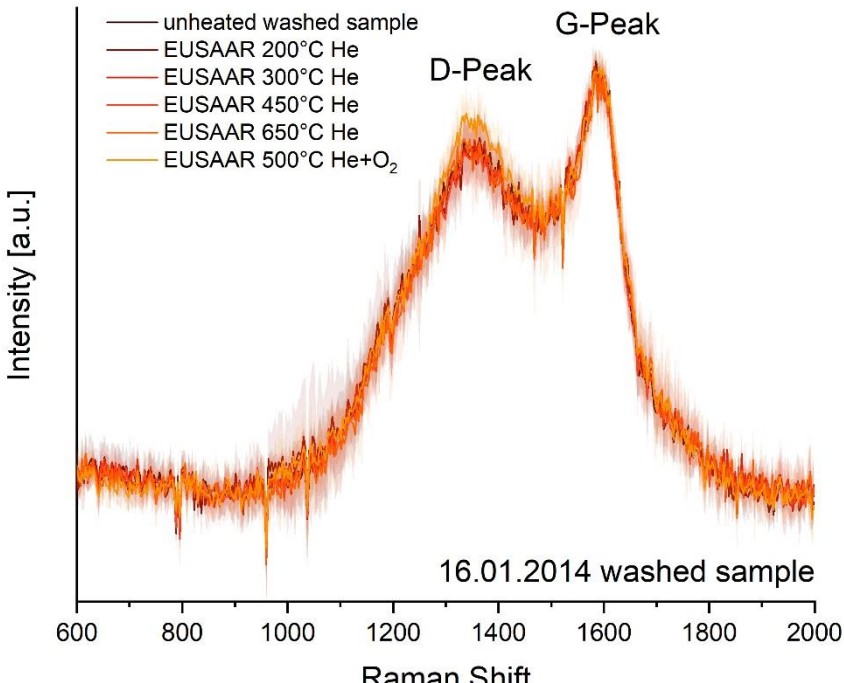

**Figure 10: Progress of the Raman spectra during the heating procedure (EUSAAR2) of a representative washed atmospheric**
**aerosol sample (16.11.2014).The Raman spectra for all washed samples show similar behavior and do not change noticeably during heating following EUSAAR2.**



## 4 Conclusion

The aim of this study was to investigate structural changes of carbonaceous matter during thermal-optical analyses. Two different thermal-optical measurement protocols with different maximum temperatures in the He phase (EUSAAR2, NIOSH870) were used to show the progress of restructuring of carbonaceous material during heating. EUSAAR2 is used in Europe as a standard method to monitor EC and OC (Brown et al., 2017). One of the arguments for the introduction of EUSAAR2 was that less PC is formed during EUSAAR2 in comparison with other (NIOSH-like) temperature protocols

(Cavalli et al., 2012). However, this argument was solely based on the decrease of the transmission laser signal. Structural changes of the material were not investigated. We found now that the D/G peak ratios of the Raman spectra can be higher for samples heated to the maximum He temperature of NIOSH870 compared to samples heated to the maximum He temperature of EUSAAR2 at least for some samples, which means that PC formed during NIOSH870 might have a higher degree of structural ordering in these cases and therefore also a different specific light absorption coefficient. The same amount of PC

would lead to a stronger decrease of the red laser signal at 635 nm for PC built during NIOSH870 compared to EUSAAR2. When comparing the Raman spectra for samples heated following the EUSAAR2 protocol we found a rather complex behaviour of structural changes. Since changes in the Raman signals (i.e. the ratio of the peak intensities) occur at different temperature levels, we defined six "Raman categories" of samples depending on the temperature levels, where visible changes in the Raman spectra occur and how fast they proceed. We tried to find similarities for samples with similar

progress of structural changes (i.e. the attribution to one of the six Raman categories) and chemical composition (BC, BrC, major ions), but did not find any relationship. This is in good agreement with Wang et al. (2010) and Yu et al. (2002) who found that non-carbonaceous components have conflicting influences of the charring behavior and the split point during thermal-optical analyses.

We also investigated the progress of the thermal-optical transmission laser signals in relation to the Raman categories

corresponding to the temperature levels where visible structural changes occur. It was obvious that the decrease of the transmission signal did not occur at the same temperature steps where structural changes in terms of ordering were found. This lack of a relation is most prominent for the samples, where the Raman spectra do not change during the whole heating procedure in He: Although the Raman spectra do not indicate a graphitization, the transmission (and reflection) laser signals still decrease during the He phase. Moreover no noticeable differences can be seen between the transmission laser signal

pattern of samples where no increase of ordering occurs in comparison with samples where the Raman spectra change during heating.

We therefore conclude that there must be processes other than graphitization also in atmospheric aerosol samples that lead to a darkening of the sample (i.e. more absorption at 635 nm) and do not affect the relative peak heights of the Raman signal (similar findings as well as their interpretation were described for the CAST soot samples in our previous publication; Haller

et al., 2019). This could be for instance the separation of oxygen and hydrogen at temperatures above 250°C in He (Petzold





et al., 2013; Chow et al., 2004) which could lead to a darkening of the material without affecting the Raman signal and/or a possible formation of light absorbing intermediate (non-graphitic) OC products as also proposed by Yu et al. (2002).

For the washed samples we found that the laser signal did not decrease during the heating procedure which is in good agreement with Yu et al. (2002) who observed that water soluble material contributed most to charring. For all washed
samples we found that the Raman signals did not change appreciably during heating in He following the EUSAAR2 protocol and we therefore infer that the relatively constant transmission signal can indeed be interpreted as a lack of graphitization.

In the existing literature the darkening of the sample is often explained by "charring" or "pyrolysis" although exact explanations of these two terms are not given. Based on our findings, it is important to point out that what is called "charring" or "pyrolysis" is not necessarily graphitization. Particularly the group of samples where the Raman signal does
not change over the whole heating procedure gives evidence that the darkening of the sample cannot be understood as graphitization in all cases. We therefore suggest that the widely used terms "charring" and "pyrolysis" should be used carefully when the darkening of a sample during thermal-optical measurement procedures is interpreted.

Summarizing we can say that restructuring of carbonaceous matter during thermal-optical measurement is very complex and not easily predictable for atmospheric aerosols. At least for the washed samples the constant transmission / reflection signal
indicates an absence of graphitization, but for all samples a decrease of the transmission or reflection laser signal during the heating procedures is not a reliable indicator for a graphitization since other processes can also lead to a darkening of the material.

*Data availability.* Data can be accessed by contacting the corresponding author.

*Author contributions.* TH performed the conceptualization of the experimental setup, most measurements (except TEM, BrC/BC, ion chromatography), the data evaluation and interpretation and prepared the manuscript. ES performed the BrC/BC measurements. TS performed the ion chromatography measurements and wrote the text in the measurement section. CR performed the TEM measurements and wrote the TEM section. AW supervised the BrC/BC measurements. AKG
contributed to discussions and provided input for the ion chromatography and to the manuscript in general. HG contributed to discussions and provided input for the Raman measurements. RH performed the conceptualization and supervision of the study, contributed to discussions, and provided extensive input to the text.

**Acknowledgements**
This work was supported by the Austrian Science Fund (FWF), grant P26040. The integrating sphere technique was developed within grant H-85/92, Hochschuljubiläumstiftung der Stadt Wien.

We thank the Analytical Instrumentation Center (AIT) of the Vienna University of Technology and the group of Prof. Bernhard Lendl for providing the Raman Microscope.



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
