# Peer review of "Investigation of structural changes of atmospheric aerosol samples during two thermal-optical measurement procedures (EUSAAR2, NIOSH870)"

_Atmospheric Measurement Techniques, 2020_

## Referee Comment (RC1) · Anonymous Referee #2 · 5 Dec 2020

This study investigated the structural changes of aerosol while heating of atmospheric aerosol samples during thermal-optical measurements. Two different thermal-optical measurement protocols (EUSAAR2, NIOSH870) were investigated. Charring of organic carbon is an important issue to investigate and can impact the thermal-optical measurements. The authors used Raman spectroscopy to track structural changes upon heating the atmospheric aerosol samples. They also used ion chromatography and integrating sphere measurements to derive ionic compositions and light absorbing fraction of carbonaceous material. They used different peak ratios derived from Ra-

man spectra as an indicator to probe structural changes. The authors observed that D/G peak ratios of the Raman spectra was higher for sample heated vis NIOSH870, suggesting higher degree of structural ordering. Samples heated via EUSAAR2 protocol exhibited complex behavior of structural changes. Investigation of washed sample with low water-soluble organic carbon and inorganic salts showed no structural ordering and darkening of sample. Overall, the authors suggested that structural ordering of sample may not be responsible for darkening of sample.

I think the research topic is relevant and important for the community. However, the overall presentation of the manuscript bit convoluted and it can be structured better way so that it will easier for readers to follow. For example, discussion of structural changes for unwashed (original) and washed sample should be discussed together. Perhaps a comparison of Fig 2 and Fig. 10, maybe just compare for one temperature would be useful for readers. Some of discussions need to elaborate. When you state about some observation and I suggest elaborating what does that mean. After revision of the manuscript, I think it will be suitable for publication.

Specific comments:

In the abstract the authors highlighted that structural ordering may not be responsible for darkening of sample, I suggest the authors to put the hypothesis here, what might be the possible cause for this.

I was wondering if the pre-edge of the spectra was considered or normalized while comparing the spectra? It was not clear to me how the different "Raman categories" were assigned? I suggest the authors to provide quantitative peak ratios to define more robust categories.

It was not clear to me how the authors relate the structural changes with the changes in the transmission laser signal. Having a quantified variable structural change (may be using the peak ratios) and compare with the transmission laser signal will be useful. I see the Fig 7 discussed the BrC/LAC ratios with qualitative structural changes. A

quantified comparison would be useful here too.

Comparison of washed and unwashed samples are interesting. Just wondering if authors investigated any standard water-soluble organic carbon to investigate the laser signal. Observation of higher initial laser signal of washed samples compared to unwashed samples need to be discussed in detail.

I was wondering about the sensitivities of the Raman peaks here. How confident authors are regarding the graphitization of carbon. Some of the previous studies showed changes in graphitic structure upon heating black carbon particles using high resolution TEM imaging.

Why the early and fast category of Raman categories are noisier compared to others?

The increase of laser signal above 650 degree C maybe due to the decomposition of the dark intermediate OC products without graphitic structures. Please elaborate the discussion about the decomposition of the dark intermediate OC products.

The authors stated that EUSAAR2 produced less pyrolyzed carbon. Please add some discussion here.

---

## Referee Comment (RC2) · Anonymous Referee #1 · 9 Dec 2020

The manuscript by Haller et al. studies atmospheric light-absorbing carbon (LAC) in particulate matter (PM) by measuring quartz filter samples with thermal-optical analysis (TOA) and Raman spectroscopy. Filter punches were removed from the TOA instrument at various temperature steps and measured with Raman spectroscopy. Filters were also "washed" by stirring in water and measured again. The filters were categorized according to "brown carbon" content as BrC/LAC and the authors discuss the interpretation of the NIOSH870 and EUSAAR2 protocols in light of their results. The experiments provide very useful data for the interpretation of LAC measurements and

warrant publication in AMT. However, the writing, analysis, interpretation, and literature discussion could all be improved.

MAJOR COMMENTS

The filter samples were categorized conceptually in Section 3.2.2 according to features shown in Figure 3. These categories are fundamentally an attempt to position the samples on the plane of D/G ratio (first axis) and TOA temperature (second axis). The authors can very simply produce a plot of D/G vs T to demonstrate their interpretation. This would address an overall lack of higher-level analysis in this manuscript. I appreciate that the authors have shown extensive amounts of raw data, but summary plots are needed.

On line 430, the authors note that the Raman spectra of washed samples do not change noticeably during TOA, even when the unwashed counterparts did change. This is the key result of this entire manuscript and should be introduced first. This experiment shows that all observed changes in Raman spectra can be attributed to PC formation (carbonization of organic compounds to form EC during heating).

It is not clear to me that the authors' criticisms of earlier work on PC are valid. On lines 450-460 the authors summarize that earlier work argued that more PC is formed during NIOSH870. The current work shows that the PC which forms during NIOSH870 has a higher degree of structural order. I agree with the conclusion but I don't agree that the two are in contradiction. "PC" is measured optically. LAC with a higher degree of graphitization will absorb more. So, it will be correctly described as "more PC". Please rephrase the conclusion here. This comment aplso applies at line 477 (and 489 and 281) where the authors claim that "processes other than graphitization lead to a darkening ... for instance the separation of oxygen and hydrogen [...] not necessarily graphitization". Yes – these processes are carbonization of organics, especially WSOC. Carbonization is, by definition, the process being described.

In general, uncertainty calculations were not described. Please refine this, and replace
statements like "outside of the error bars" with "significantly different according to a t-test".

Section 3.1, the argument about diesel cars is invalid. It is likely that residential wood burning produces a similar ratio of BC/PM to diesel cars, so the fraction of BC in PM 2.5 need not change.

At the end of Section 3.2.1 the authors should discuss possible hypotheses to explain their results. They may find inspiration from the literature, for example the similar study by Kim Cuong Le et al. 2019 https://doi.org/10.1016/j.combustflame.2019.07.037. Possibilities include the formation of PC from organics with a range of degrees of order (carbonization) and catalysis by sulfate or metals which varies between samples.

MINOR COMMENTS

The authors should probably follow Petzold et al. (2013)'s recommendation and call their Raman signals EC, not BC or LAC. (They are certainly not LAC: see comment on washing above.)

The discussion of Yu et al. (2002) on line 370 should be introduced in the introduction.

Generally, "WSOC", not "WSOCs" (applies also to the figures)

The introduction could be more structured as it moves back and forth between topics.

Line 13, "have been" not "was" found.

Line 167, please explain why the detection limit is higher for OC than EC.

Table 1, instead of grey shading add a column for Raman measurements.

Line 197, please comment on wavelength dependence of Raman measurements here.

Line 203, why was lowe rlaser power used? Please explain.

Line 205-215 please mention that 5+ Raman peaks are often fitted.

Line 224 please cite the calculation of mean crystallite size.

Line 246 please cite.

Figure 1 and others: define error bars in caption.

Please don't abbreviate the words transmittance and reflectance to trans and refl. Simply write out the words.
* * *

---

## Author Comment (AC1) · 15 Mar 2021

We re-evaluated our whole data set, checked all numbers and performed t-tests to obtain a quantitative measure for the assignment of the samples to the Raman categories. The introduction was shortened somewhat without detracting from the main message, new references were added, the text was edited and revised to address the Reviewers' comments, the order of the sections was changed (section about washed samples was shifted towards the front) and new figures (Fig. 6, additions to Fig. 3, 4 and 5)

were added to better demonstrate our findings and to give better graphical overviews. The former figures 4 and 5 were shifted to the supplementary material. In former section 2.2 (sample selection) the part about air mass back trajectories was removed because we did not use this information in the MS. A new section "sample selection and preparation" was inserted (2.3) which contains now parts of former section 2.2, section 2.1 and section 3.1. The t-tests refined the sample assignment to the Raman categories, which led to a re-assignment of two samples (30.06.2015, 03.01.2015). One sample (13.06.2015) was categorized as "non-defined" since the Raman spectra for one heated sample was measured with another exciting laser wavelength than the others in the EUSAAR2 heating series of the respective filter sample. We checked again whether only samples analyzed with the same laser for the Raman analysis were compared to each other, which resulted in the removal of 6 samples analyzed with the NIOSH870 protocol from the discussion. This reduction of samples, however, did not change our findings. The specific comments of the reviewers are addressed below.

Anonymous Reviewer 1 The manuscript by Haller et al. studies atmospheric light-absorbing carbon (LAC) in particulate matter (PM) by measuring quartz filter samples with thermal-optical analysis (TOA) and Raman spectroscopy. Filter punches were removed from the TOA instrument at various temperature steps and measured with Raman spectroscopy. Filters were also "washed" by stirring in water and measured again. The filters were categorized according to "brown carbon" content as BrC/LAC and the authors discuss the interpretation of the NIOSH870 and EUSAAR2 protocols in light of their results. The experiments provide very useful data for the interpretation of LAC measurements and warrant publication in AMT. However, the writing, analysis, interpretation, and literature discussion could all be improved.

MAJOR COMMENTS The filter samples were categorized conceptually in Section 3.2.2 according to features shown in Figure 3. These categories are fundamentally an attempt to position the samples on the plane of D/G ratio (first axis) and TOA temperature (second axis). The authors can very simply produce a plot of D/G vs T to demonstrate their interpretation. This would address an overall lack of higher-level analysis in this manuscript.

Diagrams for each category showing D/G vs T were added.

I appreciate that the authors have shown extensive amounts of raw data, but summary plots are needed.

We changed three plots to give better graphical overviews: A new Figure 6 was inserted instead of former Fig 4 and 5: Instead of showing the laser signals over the whole heating process we plotted only their values at the end of each temperature level (for the first five temperature-levels) and for comparison the D/G ratios for the same samples. This was done for each Raman category separately. In former figure 10 (now Fig. 3), which showed the Raman spectra of a washed filter, Raman spectra of the unwashed filter were added for a better visual comparison. In the plots showing examples of Raman spectra for each category (former Fig. 3) diagrams with the progress of D/G over temperature were added (now Fig. 5).

On line 430, the authors note that the Raman spectra of washed samples do not change noticeably during TOA, even when the unwashed counterparts did change. This is the key result of this entire manuscript and should be introduced first. This experiment shows that all observed changes in Raman spectra can be attributed to PC formation (carbonization of organic compounds to form EC during heating).

We shifted this chapter towards the front (now chapter 3.2) and introduced the changes of the washed and unwashed samples directly after the description of the original samples to highlight this key result.

It is not clear to me that the authors' criticisms of earlier work on PC are valid. On lines 450-460 the authors summarize that earlier work argued that more PC is formed during NIOSH870. The current work shows that the PC which forms during NIOSH870 has

a higher degree of structural order. I agree with the conclusion but I don't agree that the two are in contradiction. "PC" is measured optically. LAC with a higher degree of graphitization will absorb more. So, it will be correctly described as "more PC". Please rephrase the conclusion here.

We removed this argument and rephrased the whole paragraph, which now reads: "These differences could be caused by a stronger graphitization during the NIOSH870 protocol than during EUSAAR2 at least in three cases out of six. This would mean that PC developed during the heating with different protocols might also have different optical properties which would influence the decrease of the transmission / reflection laser signal during the thermal-optical measurement procedure. This would be in good agreement with Yu et al. (2002), who found that light absorption coefficients of PC formed during a thermal-optical heating procedure are not identical over the whole heating procedure. It is plausible that this might be true also for two protocols with different maximum temperatures in He. The higher D/G ratio for a higher heating temperature is also in accordance with Le et al. (2019) who found increasing D/G ratios between 600°C and 800°C for their OC rich samples heated in N2. As suggested by Chow et al. (2001) part of EC could leave the filter already during the 870°C temperature level in helium because of oxygenation or catalysis caused by mineral or other oxides at temperatures >700°C. This premature evolution of relatively structured material could also affect the D/G ratio during the highest inert temperature step in NIOSH870. Therefore we assume that the higher D/G ratio of NI870 for some samples could be a combination of stronger graphitization caused by the higher temperature and premature oxidation of EC during the 870°C level in NIOSH870."

This comment also applies at line 477 (and 489 and 281) where the authors claim that "processes other than graphitization lead to a darkening ... for instance the separation of oxygen and hydrogen [...] not necessarily graphitization". Yes – these processes are carbonization of organics, especially WSOC. Carbonization is, by definition, the process being described.

We changed the term "graphitization" to "increase of structural ordering" in this section and in the conclusion (Lines 434, 438).

In general, uncertainty calculations were not described. Please refine this, and replace statements like "outside of the error bars" with "significantly different according to a t-test".

T-tests were performed to show which changes in the Raman spectra were significant and which were not. The t-tests confirmed the assignments to the Raman categories for all samples except for two: One sample (03.01.2015) was assigned to the category "late and fast" instead of "non-defined" after the t-test. For the second sample (30.06.2015) which we had assigned to the category "early and fast" the p value of the t-test was just above the significance level. Therefore we shifted this sample to the category "non-defined". The respective graphics were changed.

Section 3.1, the argument about diesel cars is invalid. It is likely that residential wood burning produces a similar ratio of BC/PM to diesel cars, so the fraction of BC in PM 2.5 need not change.

We refined this argument and included data about domestic space heating and fuel use in Vienna. Biomass burning is used by less than 1

At the end of Section 3.2.1 the authors should discuss possible hypotheses to explain their results. They may find inspiration from the literature, for example the similar study by Kim Cuong Le et al. 2019 https://doi.org/10.1016/j.combustflame.2019.07.037. Possibilities include the formation of PC from organics with a range of degrees of order (carbonization) and catalysis by sulfate or metals which varies between samples.

We broadened the discussion in section 3.2.1 (now section 3.3.1) and referred also to the paper by Le et al. (2019) who found increasing D/G ratios also between 600°C and 800°C and give therefore evidence that the structural ordering at the end of the inert phase of the NIOSH870 protocol could indeed be higher than at the end of EUSAAR2.

We also point to the possibility of premature oxidation of PC or EC during the highest temperature level in NIOSH870 as suggested by Chow et al. (2004). The reference to Le et al. (2019) was also inserted in section 3.3.3 for a discussion of the decreasing laser signal without increasing structural ordering which could be caused by a removal of C-H "out-of-plane" bonds around 350-400°C, a decomposition of surface carbonyl (C-O, C=O) groups or a decrease of cross-linkages between polyaromatic units and carbon chains (Le et al., 2019).

MINOR COMMENTS The authors should probably follow Petzold et al. (2013)'s recommendation and call their Raman signals EC, not BC or LAC. (They are certainly not LAC: see comment on washing above.)

We do not call our Raman signals BC or LAC. The Raman spectra are coded only with the date of the sampling day, as well as the heating temperature and measurement protocol. The samples were then categorized according to changes in the Raman spectra and these categories were compared to several features of the original samples, among others also BrC/LAC. EC values are not derived from the Raman spectra in our work. In our nomenclature, we followed Petzold et al. (2013) already in the first version of the MS: We call results obtained by thermal-optical analysis EC, OC or TC, and results obtained by integrating sphere measurements BrC, BC or LAC.

The discussion of Yu et al. (2002) on line 370 should be introduced in the introduction.

We revised the discussion about processes occurring during heating that lead to a darkening of the sample but not to an increase of structural ordering. Our main reference for this is now the paper by Le et al. (2019) who give more detailed suggestions on the processes happening at lower temperatures in the absence of oxygen. Both references, Yu et al (2002) as well as Le et al. (2019), are discussed in the new introduction.

Generally, "WSOC", not "WSOCs" (applies also to the figures)

done

The introduction could be more structured as it moves back and forth between topics.

We changed the structure of the introduction and eliminated some repetitions.

Line 13, "have been" not "was" found.

done

Line 167, please explain why the detection limit is higher for OC than EC.

Actually, OC and EC have the same detection limit of $0.1\mu gC/cm^2$. We corrected this in the MS.

Table 1, instead of grey shading add a column for Raman measurements.

A new column was inserted showing whether Raman measurements were performed for the respective temperature level and the shading was removed.

Line 197, please comment on wavelength dependence of Raman measurements here.

A short description of the wavelength dependence of D/G ratio was added in line 171. Since we use the same excitation laser for all measurements done on a single sample and compare only the relative changes among them, the wavelength dependence does not interfere in our analyses.

Line 203, why was lower laser power used? Please explain.

An explanation was added in line 151: "Generally laser power was lowered for highly absorbing samples to prevent thermal destruction of the material, and for samples containing fluorescent material to reduce the interfering fluorescence background."

Line 205-215 please mention that 5+ Raman peaks are often fitted.

We now mention that several authors fit 5+ Raman peaks to their soot spectra and added the following references: Le et al. (2019), Sadezky et al. (2005), Zickler et al.

(2006).

Line 224 please cite the calculation of mean crystallite size.

For the calculation of the mean crystallite sizes the Scherrer equation was used. The reference (Fultz, B. and Howe, J. M.: Transmission electron microscopy and diffractometry of materials, Physics and astronomy online library, Springer, Berlin, 2001.) was added.

Line 246 please cite.

These data were obtained in our study.

Figure 1 and others: define error bars in caption.

done

Please don't abbreviate the words transmittance and reflectance to trans and refl. Simply write out the words.

done

Please also note the supplement to this comment:
https://amt.copernicus.org/preprints/amt-2020-398/amt-2020-398-AC1-
supplement.pdf

---

## Author Comment (AC2) · 15 Mar 2021

We re-evaluated our whole data set, checked all numbers and performed t-tests to obtain a quantitative measure for the assignment of the samples to the Raman categories. The introduction was shortened somewhat without detracting from the main message, new references were added, the text was edited and revised to address the Reviewers' comments, the order of the sections was changed (section about washed samples was shifted towards the front) and new figures (Fig. 6, additions to Fig. 3, 4 and 5)
AMTD

were added to better demonstrate our findings and to give better graphical overviews. The former figures 4 and 5 were shifted to the supplementary material. In former section 2.2 (sample selection) the part about air mass back trajectories was removed because we did not use this information in the MS. A new section "sample selection and preparation" was inserted (2.3) which contains now parts of former section 2.2, section 2.1 and section 3.1. The t-tests refined the sample assignment to the Raman categories, which led to a re-assignment of two samples (30.06.2015, 03.01.2015). One sample (13.06.2015) was categorized as "non-defined" since the Raman spectra for one heated sample was measured with another exciting laser wavelength than the others in the EUSAAR2 heating series of the respective filter sample. We checked again whether only samples analyzed with the same laser for the Raman analysis were compared to each other, which resulted in the removal of 6 samples analyzed with the NIOSH870 protocol from the discussion. This reduction of samples, however, did not change our findings. The specific comments of the reviewers are addressed below.

Anonymous Reviewer #2 This study investigated the structural changes of aerosol while heating of atmospheric aerosol samples during thermal-optical measurements. Two different thermal-optical measurement protocols (EUSAAR2, NIOSH870) were investigated. Charring of organic carbon is an important issue to investigate and can impact the thermal-optical measurements. The authors used Raman spectroscopy to track structural changes upon heating the atmospheric aerosol samples. They also used ion chromatography and integrating sphere measurements to derive ionic compositions and light absorbing fraction of carbonaceous material. They used different peak ratios derived from Ra- man spectra as an indicator to probe structural changes. The authors observed that D/G peak ratios of the Raman spectra was higher for sample heated vis NIOSH870, suggesting higher degree of structural ordering. Samples heated via EUSAAR2 protocol exhibited complex behavior of structural changes. Investigation of washed sample with low water-soluble organic carbon and inorganic salts showed no structural ordering and darkening of sample. Overall, the authors suggested that structural ordering of sample may not be

responsible for darkening of sample. I think the research topic is relevant and important for the community. However, the overall presentation of the manuscript bit convoluted and it can be structured better way so that it will easier for readers to follow. For example, discussion of structural changes for unwashed (original) and washed sample should be discussed together. Perhaps a comparison of Fig 2 and Fig. 10, maybe just compare for one temperature would be useful for readers.

We replaced figure 10 with a figure comparing the Raman spectra obtained for the same filter where one part was washed and the other unwashed. Washed samples and their unwashed counterparts, were analyzed only for the EUSAAR2 protocol, because one half of each filter was used for washing so no material was left for additional analysis with NIOSH870. A direct comparison between Fig. 2 and Fig. 10 is not feasible, because Fig. 2 shows a comparison between spectra obtained for samples heated according to the NIOSH870 and EUSAAR2protocols.

Some of discussions need to elaborate. When you state about some observation and I suggest elaborating what does that mean. After revision of the manuscript, I think it will be suitable for publication.

Specific comments: In the abstract the authors highlighted that structural ordering may not be responsible for darkening of sample, I suggest the authors to put the hypothesis here, what might be the possible cause for this. We added our hypothesis at the end of the abstract: "Possible transformations at lower temperatures could include the formation of non-graphitic light absorbing intermediate organic carbon, the release of C-H groups or the decrease of carbonyl groups."

I was wondering if the pre-edge of the spectra was considered or normalized while comparing the spectra?

The spectra were normalized to the maximum of the G-Peak after subtracting the background, which differed for different samples because of the various amounts of light absorbing species. The normalization to the G-peak accounts for these variations and

was therefore preferred over a pre- or near-edge normalization.

It was not clear to me how the different "Raman categories" were assigned? I suggest the authors to provide quantitative peak ratios to define more robust categories.

We performed t-tests to substantiate the assignment of the samples to the categories and added diagrams showing the change of the peak ratios with increasing temperature.

It was not clear to me how the authors relate the structural changes with the changes in the transmission laser signal. Having a quantified variable structural change (maybe using the peak ratios) and compare with the transmission laser signal will be useful.

Diagrams were added showing the change of the D/G ratio compared to the change of the laser signal for each sample. (Fig. 6)

I see the Fig 7 discussed the BrC/LAC ratios with qualitative structural changes. A quantified comparison would be useful here too.

We quantified the categorization of the changes of the Raman spectra by performing a t-test. So BrC/LAC ratios are now compared with mathematically more stable categories of structural changes.

Comparison of washed and unwashed samples are interesting. Just wondering if authors investigated any standard water-soluble organic carbon to investigate the laser signal.

An earlier analysis with sucrose showed that transmission and reflection laser signals decreased during TO measurements and that the pyrolyzed material showed the two typical soot peaks as well. The investigation of charring of sucrose, however, is outside the scope of this study, which is focused on atmospheric samples.

Observation of higher initial laser signal of washed samples compared to unwashed samples need to be discussed in detail.

Since the washed samples were mechanically shaken, also part of BC was removed from the filter. Integrating sphere measurements showed that absolute BC loadings were lower after washing. We added a sentence in line 255 to state this more precisely.

I was wondering about the sensitivities of the Raman peaks here. How confident authors are regarding the graphitization of carbon. Some of the previous studies showed changes in graphitic structure upon heating black carbon particles using high resolution TEM imaging.

We are confident that the sensitivity of the Raman peaks is acceptable for the purpose of this study. In our recent study (Haller et al., 2019) we used TEM imaging and electron scattering additional to Raman spectroscopy for the investigation of structural changes of miniCAST-soot (i.e. propane combustion aerosol; sootgenerator.com). The Raman spectra as well as the electron diffraction patterns did not change for a sample with a large amount of BC. The advantage of Raman spectroscopy was that we could get averaged information about the whole illuminated area and that the quartz fiber filter did not influence the measurements. Our experiences with TEM showed that the electron beam interfered with the quartz fibers, which made the sample preparation and the measurement very complicated.

Why the early and fast category of Raman categories are noisier compared to others?

The signal to noise ratios as well as the signal to fluorescence background differed for different samples. The first occurred because of differing absorbance depending e.g. on the amount of BC on the filter and hence a generally lower Raman signal. The latter might be caused by the presence of fluorescent species (An analysis of these species was beyond the scope of the present study.)

The increase of laser signal above 650 degree C maybe due to the decomposition of the dark intermediate OC products without graphitic structures. Please elaborate the discussion about the decomposition of the dark intermediate OC products.

We changed the whole paragraph to clarify our argument about building of non-graphitic structures – which is the main message of it – and deleted the sentence about decomposition of dark intermediate OC products. After the changes this additional information is no more relevant at this place.

The authors stated that EUSAAR2 produced less pyrolyzed carbon. Please add some discussion here. This statement was made by Cavalli et al. (2010) who found from the analysis of the transmission laser signal that more PC was built during a NIOSH protocol compared to EUSAAR protocol. We added the reference a second time in the introduction (line 54) for clarification.

Please also note the supplement to this comment:
https://amt.copernicus.org/preprints/amt-2020-398/amt-2020-398-AC2-supplement.pdf